# How Can Ozone and Relative Humidity Affect Artists’ Alkyd Paints? A FT-IR and Py-GC/MS Systematic Study

**DOI:** 10.3390/polym14091831

**Published:** 2022-04-29

**Authors:** Laura Pagnin, Elisabetta Zendri, Francesca Caterina Izzo

**Affiliations:** 1Academy of Fine Arts Vienna, Institute of Science and Technology in Art, Schillerplatz 3, 1010 Vienna, Austria; l.pagnin@akbild.ac.at; 2Department of Environmental Sciences, Informatics, and Statistics, Ca’ Foscari University of Venice, Via Torino 155/b, 30174 Venice, Italy; elizen@unive.it

**Keywords:** alkyd paints, Py–GC/MS, ozone, relative humidity, degradation behaviour

## Abstract

Knowledge of the chemical–physical reactions that determine the main degradation behaviour of artists’ alkyd paints represents one of the main problems within the museum exhibitions. The collection and interpretation of these data on degradation phenomena, especially after ozone exposure at different relative humidity values, can be useful for their conservation needs. Therefore, a systematic investigation of these materials may help achieve this goal. Firstly, surface-level identification of the main functional groups of ad hoc created and aged alkyd paints was performed using attenuated total reflection Fourier-transform infrared spectroscopy (ATR-FTIR). Subsequently, these paints were investigated by pyrolysis–gas chromatography and mass spectrometry (Py–GC/MS), allowing for precise bulk identification of the organic compounds before and after accelerated ageing. A first successful attempt to provide quantitative Py–GC/MS data on alkyd-based paints is here presented and discussed. Comparing the results, it was possible to obtain new insights into the degradation behaviour of alkyd paints when exposed to ozone, allowing us to devise specific preventive and conservation strategies for these artistic materials.

## 1. Introduction

The 20th century represents the historical period of maximum development and innovation of new pigments and polymeric binders used in the artistic and industrial fields. The new formulations have led artists to develop new artistic techniques and renewal of artworks both from a conceptual and material point of view [1]. 

In the late 1930s, the paint industry developed so-called alkyd resins (term deriving from alcohol and organic acids) as an evolution of oil paints; their introduction was intended to achieve similar optical properties, but with a lower cost and greater versatility in formulations. Alkyd paints are condensation products of polybasic acids with poly-functional alcohols, i.e., polyesters, co-esterified with drying oils (such as linseed and sunflower oil) [2]. The drying and curing behaviour of alkyd resins involves complex oxidative polymerization reactions resulting in a cross-linked and insoluble film. Considering that the polyester component in the binder is already cross-linked, the necessary drying reactions are less than in an oily binder and therefore have a faster drying rate [3,4]. 

Since their introduction, many artists have decided to employ alkyd paints, including Pablo Picasso, Peter Blake, Jackson Pollock, and Roy Lichtenstein. For this reason, the study of the degradation processes related to this binder is still of current interest and under investigation [5]. While the photodegradation reactions involving alkyd polymers have been investigated in depth [6,7,8,9], less information is present about the deterioration processes related to the interaction between this modern synthetic material and atmospheric pollutant gases. As previously verified [10], various atmospheric constituents (such as SO_2_ and NO_x_) affect the chemical stability of this polymer, mainly due to their enhanced reactivity when combined with moisture and their corrosive effect on paint surfaces. 

Among the atmospheric gases, ozone is one of the most oxidizing pollutants present in nature and has the highest concentration values in urban areas [11]. Its corrosive effects have already been tested on a variety of art materials, such as metals [12], stones, and watercolours [13], showing that its degrading action does not only change depending on the pollutant agent/material interaction, but also on its combination with sunlight, oxygen, and other pollutants. Some recent studies [14] began to investigate the degradation reactions between ozone and modern paints, demonstrating that the chemical identification of the materials under examination and the study of the degradation processes related to gaseous pollutants is a wide topic that has still to be fully investigated and elucidated. Since these pollutants are present not only in outdoor environments but also in indoor museum exhibitions [15], the different degradation paths require further research, both through use of surface spectroscopic and analytical separation techniques. 

For this reason, the focus of the present study was to understand the impact of ozone (O_3_) on alkyd paints. The painted films examined were obtained by mixing a commercial alkyd-based resin with nine inorganic pigments. The aim was to develop an analytical methodology for the investigation of the possible degradation processes developed in alkyd paints after (a relatively strong) artificial exposure to ozone at two different % relative humidity values (50 and 80%RH) for a total exposure time of 168 h. 

Since it was essential to understand what occurred to these painted films both at the superficial and bulk level, attenuated total reflection Fourier-transform infrared spectroscopy (ATR-FTIR) and pyrolysis–gas chromatography/mass spectrometry (Py-GC/MS) were employed, respectively. 

FTIR is generally and widely used to identify alkyd paint formulations by detecting the typical absorption bands of synthetic polymers, pigments, and additives. However, the overlap of the absorption bands of polymers with those of other materials or functional groups might make its identification difficult [16]. 

In recent decades, Py–GC/MS has been extensively used to study organic materials in art, archaeology, and conservation practice, with special attention to the characterization of synthetic polymeric matrices [17,18,19]. Py–GC/MS is based on the thermal breakdown of the macromolecules of the samples, whose fragments are separated by gas chromatography and finally detected by mass spectrometry. Considering the numerous monomers of which a synthetic polymeric material can be composed, Py–GC/MS allows for their precise identification. Being a highly sensitive technique, it requires a minimum amount of sample to obtain a differentiated result based on the fragments detected and present in the complex polymeric materials. This technique was tested on various synthetic modern art materials constituted by high molecular weight molecules (e.g., poly-acrylates and poly (vinyl acetate)s) [20] and low molecular weight molecules (e.g., phenolic resins, ketone resins, hydrocarbon resins) [21]. 

Since alkyd resins have polar and low volatile compounds in their formulations (such as fatty acids from the drying oils), these polymeric fragments can show resolution problems on a non-polar column; therefore, so-called thermally-assisted methylation (THM) can be an alternative [22]. In this study, pre-treatment with TMAH (tetramethylammonium hydroxide) of the samples was chosen before pyrolysis. Upon heating, TMAH induces hydrolysis and leads to the methylation of acid compounds. With this analytical process, a better chromatographic separation is obtained. THM–Py–GC/MS is a very sensitive and reproducible technique, already successfully employed to identify alkyd resins and drying oils [23,24,25]. 

This paper presents a novelty from the point of view of the Py–GC/MS quantitative study of complex polymers subjected to degradation; to the best of our knowledge, it represents the first attempt to provide quantitative results to better elucidate the effects of ozone-induced degradation on alkyd paints. 

From the evaluation of the results obtained, suitable conservation practices could be conceived, as the degradation processes will differ depending on the pollutant agent in contact with the artworks [26]. Furthermore, by monitoring the pollutants present in an indoor or outdoor environment (in this case, focusing on ozone), it will be possible to develop an adequate preventive strategy able to protect the most sensitive artistic objects.

## 2. Materials and Methods

### 2.1. Sample Preparation

Several mock-ups were prepared by mixing pure Alkyd Medium 4 (Lukas^®^, Leipzig, Germany) and nine different inorganic pigments (Kremer Pigmente, Aichstetten, Germany). The pigment/binding medium (P/BM) ratio chosen was 1:3 by weight. This weight ratio was chosen both to obtain a homogeneous and consistent mixture (similar to the commercial formulations) and to allow an investigation of the paints’ degradation behaviours based on equal P/BM ratios. After mixing them with a ceramic mortar, paints were cast on glass slides with a film thickness of 150 μm using the so-called Doctor-Blade procedure [27]. The samples were allowed to dry at ambient conditions (approx. 22 °C and 30%RH) for three weeks. The total number of mock-ups was 30 (including a pure alkyd sample in the sets). The paint materials investigated are listed in Table 1.

### 2.2. Weathering Experiments

The ageing chamber used (Bel-Art™SP Scienceware™, Vienna, Austria) is made of a co-polyester glass (Purastar^®^, Hanoi, Vietnam), including gas inlets and outlets able to reach a total volume of 30 cm^3^. The weathering system connected to the chamber consists of a system able to mix the synthetic air with corrosive gases. Synthetic air 5.0 (Messer, Vienna, Austria) was humidified using double-distilled water and mixed with the selected gas. Subsequently, the chamber was continuously flushed with the gas mixture with a gas flow rate of 100 L/h. The relative humidity (RH) values chosen were 50 and 80%, whereas the ozone (O_3_) concentration selected was 2500 ppm. The samples were aged for 168 h. The samples renamed Reference were subjected to natural ageing; once prepared and dried, they were stored in a desiccator without light and with stable RH% and T values (35% and 20 °C, respectively). The gas concentration was monitored daily during the artificial ageing using a specific sensor for O_3_ detection (Aeroqual Limited, Auckland, New Zealand, model AQL S200). During the ageing experiments, the values could vary by ±10–13 ppm. The values of the gas concentrations were selected to reproduce accelerated outdoor ageing. According to European Environmental Agency for air quality monitoring, the annual mean concentration of O_3_ in the atmosphere is approximately 40 ppm [28]. Considering the value of the experimental concentration chosen and the average annual monitored in the atmosphere, the exposure time, deriving from the accelerated artificial ageing carried out, was equal to an approximate value of 1 year. The sample sets were named according to the ageing modalities as follow: Alk_ref, Alk_50%RHO_3_, Alk_80%RHO_3_ (for pure resin), and Alk_PW6_ref, Alk_PW6_50%RHO_3_, and Alk_PW6_80%RHO_3_ (for each pigment).

### 2.3. Attenuated Total Reflection Fourier-Transform Infrared Spectroscopy (ATR-FTIR)

For the ATR-FTIR investigations, a LUMOS FTIR microscope (Bruker Optics, Ettlingen, Germany) was used in ATR mode combined with a germanium crystal. The instrument was equipped with a cooled photoconductor MCT detector. In each sample, five measurement points were acquired in the spectral range between 4000 and 480 cm^−1^ by performing 64 scans with a resolution of 4 cm^−1^. The ATR-FTIR measurements were able to obtain chemical depth information around 0.65 μm, considering the germanium crystal refractive index (n_1_) of 4.01 and the angle of incidence (θ) of the IR beam of 45°. The resulting spectra were collected and evaluated by OPUS^®^ 8.0 software (Bruker Optics, Ettlingen, Germany). As shown in a previous study [29], to better investigate the influence that each inorganic pigment and ageing condition has on the degradation of the alkyd resin, a semi-quantitative evaluation by integrating specific IR bands was evaluated. For this data treatment, the spectra were firstly averaged, baseline corrected, and vector normalized in the 1089–1050 cm^−1^ region. This normalization region was chosen as it does not change with gas ageing. In this study, the carbonyl group C=O band (at 1719 cm^−1^, integration range from 1790 to 1615 cm^−1^) was integrated, and the corresponding area value was calculated for each aged sample. This specific band was selected because it shows strong intensity, does not overlap with other bands, and is the most representative band of the binder [30].

### 2.4. Thermally Assisted Hydrolysis and Methylation (THM)–Single Shot Pyrolysis–Gas Chromatography/Mass Spectrometry (TMH–SS-Py–GC/MS)

#### 2.4.1. Sample Treatment, Qualitative and Quantitative Analysis

Small aliquots of samples taken from the pure alkyd resin and the alkyd paints in mixture with the nine inorganic pigments were accurately weighted using a microanalytical balance (0.001 mg) and placed in eco-cup pyrolysis crucibles. The weights of the samples generally varied between 60 and 80 µg. The samples were then treated with 5 µL of tetramethylammonium hydroxide (TMAH), 25%, in methanol. As said before, TMH–Py–GC/MS with TMAH derivatization has already been proven to be a suitable treatment to obtain reproducible results on artists’ drying oils and alkyd paints [8,31].

Moreover, qualitative interpretation of obtained data, a successful attempt to quantify the main compounds detected in the pure alkyd resin and the alkyd paints (the poly-oil, polyacid, and lipidic fractions) was presented. To the best of our knowledge, this is the first attempt in the specific study of commercial alkyd paintings. Although some authors reported many problems in quantitation due to the pyrolytic procedure and pathways [32], or differences in the thicknesses of the samples taken from the paint surface [9], in the literature, there are also several studies on polymeric cultural heritage and industrial material where Py–GC/MS data is considered quantitatively [33,34,35]. To overcome reproducibility problems, the present study paid particular attention: (i) during the preparation of the samples to the thickness of the painted films (150 µm for all paint layers); (ii) during the sampling of the samples, to their weight, and (iii) during reaction with TMAH to the methylation efficiency. This issue was solved by using the internal standard procedure for the lipidic fraction and calibration curves of the most abundant compounds. Then 5 µL of a 1 mmol/mL FA-C19 solution in methanol was added to each crucible before placing it in the pyrolizer. According to a well-developed and validated methodology, this fatty acid (very similar to palmitic and stearic acids, generally unaffected by ageing) can serve as an internal standard for the quantification of fatty acids [36,37,38]. In this way, it was possible to carry out a reproducible method to quantitatively evaluate the products deriving from gaseous degradation of the lipidic fraction. After the determination of the response factors of the most abundant fatty acid in the oil fraction (palmitic, stearic, oleic, linoleic, azelaic, suberic, and sebacic acids) and their concentration in weight (after samples weight correction), the most significant molar ratios to compare oxidation induced by ozone in the presence of relative 50% and 80%RH were calculated, namely, A/P (azelaic to palmitic acids ratio), D/P (ratio of the sum of the dicarboxylic acids—azelaic, suberic, sebacic—to palmitic acid) to provide information on the degree of oxidation of oil and the total percentage of dicarboxylic acids D% (azelaic + suberic + sebacic acids), and the O/S ratio (oleic to stearic acids ratio), which can indicate the maturity of oils [24,39]. Furthermore, the P/S (palmitic to stearic acids ratio) was considered, since it is commonly used to suggest the type of drying oil [40], although it has been proven that for modern and contemporary lipidic-based binders, the traditional distinction according to P/S ratios are not always reliable [37,41].

In addition to the fatty acids present in the lipid fraction, also the % of benzoic acid (BA%), pentaerythritol (PE% as the sum of tetra tri- and di-derivatives), and phthalic acid (PhA%) were calculated, using calibration curves. According to [6,31], the ratios PhA/A (phthalic acid to azelaic acid) and PhA/P (phthalic acid to palmitic acid) were also calculated. In this study, other significative ratios were introduced to widen the understanding of possible changes occurring during ozone-induced ageing: BA/P (benzoic acid to palmitic acid), PhA/PE phthalic acid-to pentaerythritol), BA/PE (benzoic acid-to pentaerythritol), and PE/oil (polyol/oil ratio, where oil is the total % of fatty acids).

#### 2.4.2. Apparatus and Methodology

All samples were investigated with thermally assisted hydrolysis and methylation single shot Py–GC/MS (THM–SS-Py–GC/MS). The apparatus consisted of a PY-3030D pyrolizer (Frontier Lab, Koriyama, Japan) mounted on a Trace 1310 gas chromatograph (ThermoFisher Scientific, Waltham, MA, USA) combined with an ISQ7000 mass spectrometer (ThermoFisher Scientific, Waltham, MA, USA). The SS-Py–GC/MS analyses were performed at 650 °C for 0.20 min. The interface temperature was 280 °C. The interface temperature was 280 °C. The pyrolysis unit was linked by a programmed temperature vaporization (PTV) injector to a Supelco SLB5 MS column (with a length of 30 m, an internal diameter of 0.25 mm, and a film thickness of 0.25 μm). Helium was used as carrier gas (constant flow at 1 mL/min). The temperature program was set as follows: 35 °C (held 1 min)–30 °C/min–110 °C–15 °C/min–240 °C–5 °C/min–315 °C (held 2 min). The column was directly coupled to the ion source of the mass spectrometer. The interface temperature was 280 °C, and the ion source temperature was 300 °C. The transfer line to the MS was kept at 280 °C. MS detection was carried out with electron impact ionization (70 eV) in full scan mode, in the range 29–600 *m*/*z*, with 0.2 s of dwell time. The solvent delay was set at 4.30 min. For collecting and processing mass spectral data, Chromeleon 7 software was used. The THM–SS-Py–GC/MS and TD results were interpreted with NIST Libraries, F-Search software, and ADMIS ad hoc created libraries. Furthermore, the ESCAPE system, an expert system for characterizing Py–GC/MS data using AMDIS and Excel, was used [42].

## 3. Results

### 3.1. Pure Alkyd Resin

The ATR-FTIR analysis performed on pure alkyd resin and paints allowed the most characteristic functional groups related to the binder and pigments used to be identified (Table 2) [9,23,43,44,45]. 

According to the commercial datasheet, the medium used is an oil-modified polyester resin. The strong band at 1719 cm^−1^ is due to the C=O stretching vibration, common to both oil and ortho-phthalic acid or anhydride, whereas the C–O–C stretching (1258 and 1119 cm^−1^) and the aromatic in-plane and out-of-plane bending (1600–1580, 1071, and 741–711 cm^−1^) are typical for phthalic-based compound [46]. Relative to the oil portion, the CH_2_ and CH_3_ symmetric and asymmetric stretching and bending are assigned [30]. By comparing the spectra of fresh and Alk_ref (dried), the functional groups corresponding to the phthalate plasticizer were detectable. The small band at 3008 cm^−1^ was assigned to the vinyl proton of the C–H stretching, the two peaks at 1600 and 1580 cm^−1^ to the aromatic ring C=C stretching, and the peak at 768 cm^−1^ to the aromatic C–H out-of-plane bending [5]. Comparing the fresh and unaged sample spectra, the peaks at 3008 and 768 cm^−1^ were no longer detectable due to the subsequent drying stage.

To further identify the resin used in this study and better understand the behaviour of the painted layers made with it, Py–GC/MS analyses were conducted (as described in materials and methods). Using this technique, it was possible to determine the type of polyhydric alcohol and polybasic acid present in the commercial formulation. In addition, more information was obtained about the oil added to the alkyd resin and several additives. Table 3 lists all the compounds detected with their retention times and significant *m*/*z* values.

The total ion current (TIC) pyrogram obtained after THM–SS-Py–GC/MS analysis of the Alk_ref is illustrated in Figure 1. Among the numerous peaks identified, the most meaningful findings were:Thanks to the detection of phthalic acid (as dimethyl ester), the polybasic acid used in alkyd manufacture was identified as orthophthalic acid (PA). To confirm its presence and exclude other polybasic acidic structures (such as phthalic anhydride, which is the most common type of dibasic acid used in alkyd paints), thermal desorption (TD) of Alk_ref was performed (see TD in Appendix B, Figure A1). This result differed from similar studies carried out on the same type of alkyd resin Medium 4 by Lukas in 2013, where the polybasic acid was found to be phthalic anhydride [6]. However, it is also known that industrial formulations may vary in composition/raw materials for their preparation, so this new variation is not entirely unexpected; in 2016 Anghelone et al. [9] revealed the presence of orthophthalic-based resin.The polyol was identified as pentaerythritol (PE), a five-carbon tetraol commonly used since the 1960s in the resin formulation. Upon TMAH derivatization and THM–SS-Py–GC/MS analysis, PE dissociated into di-, tri-, and tetra-PE methyl esters. Compared to previous publications on the same commercial resin [9], the % of polyol in Alk_ref film was higher. Lukas alkyd was found to contain between 8 and 21% in fresh and aged paints by Wei et al. [6], whereas in the current study, the PE% was 29% (see Table 4). As seen above, this difference was not mainly due to analytical methodologies but likely to a different blend within the commercial formulation. Moreover, other studies reported very dissimilar contents of PE (ranging from 23% to 0%) according to the blend considered and the presence of organic and inorganic pigments [31].The detection of saturated monocarboxylic acids (palmitic and stearic acids being the most abundant), dicarboxylic saturated acids (suberic, azelaic, sebacic acids), and unsaturated fatty acids (oleic and linoleic acids) allowed the presence of a vegetable drying oil to be identified. According to the calculated molar ratio between palmitic and stearic acids (P/S = 1.6), the oil component could be linseed oil. Nevertheless, as mentioned before, other kinds of drying or semi-drying oils might be present. The amount of oil (summing the most abundant fatty acids) of the Alk_ref was about 30%, as reported in Table 4.One of the most significant peaks was related to benzoic acid (BA), an aromatic monobasic acid generally present in Py–GC/MS alkyd data. Considering the relatively high % of BA (27%, Table 4), it is more likely present as a stopping agent in the commercial formulation more than a pyrolysis product of the polybasic acid.Several additives were detected, in particular phthalate-based compounds (such as nonyl phthalate, octyl phthalate, furfuryl hexyl phthalate) commonly added to the polymeric mixtures as plasticizers. A paint stabilizer, a UV-light absorbing piperidine compound, namely 4-methyl-2-piperidone, was identified at 6.751 min [47]. At 10.774 min, the presence of butanal dimethylhydrazone is related to its use as a paint drier [48]. Figure 2 illustrates the proposed polycondensation reaction, starting from the main compounds identified in PY–GC/MS analysis.

**Figure 1 polymers-14-01831-f001:**
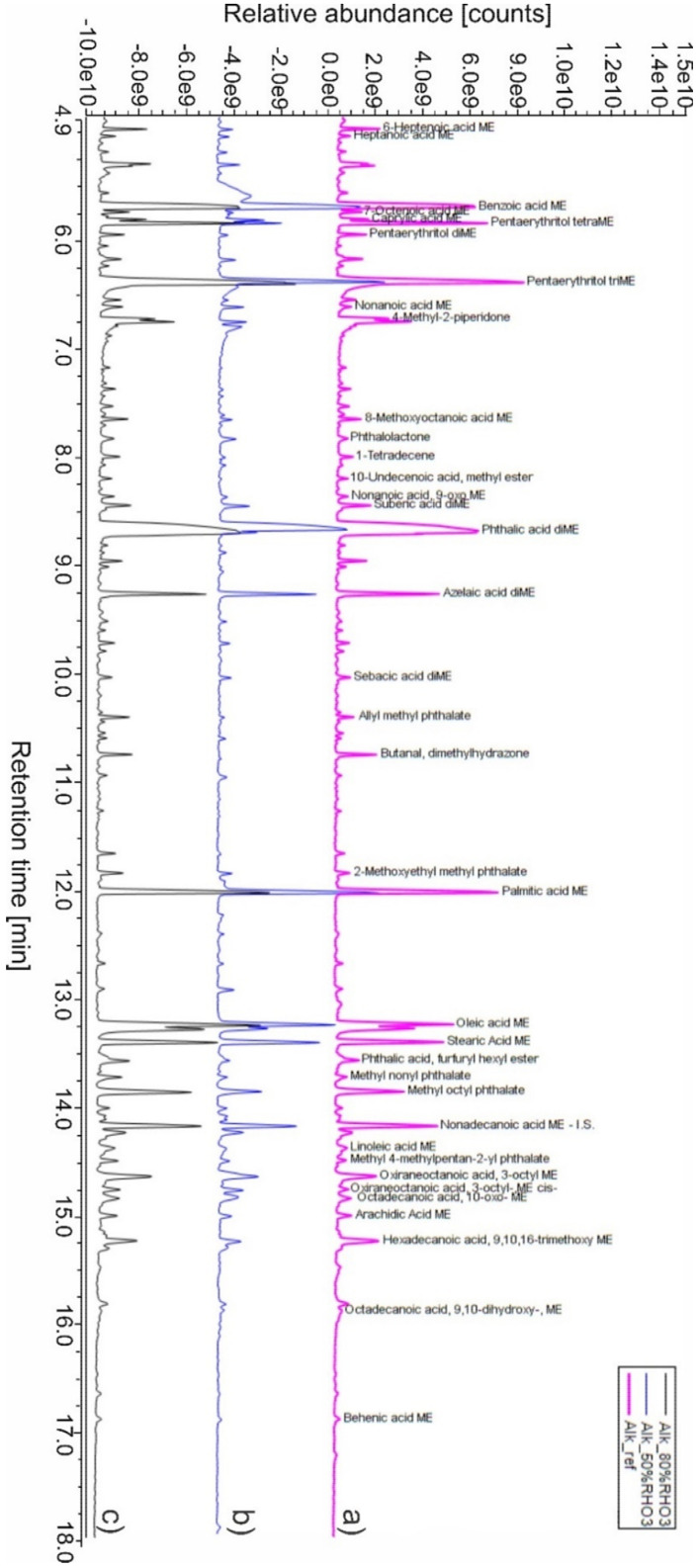
Total ion current (TIC) pyrograms obtained from (**a**) Alk_ref, (**b**) Alk_50%RH, and (**c**) Alk_80%RH samples.

After ozone ageing, some meaningful observations were shown by FT-IR and Py–GC/MS. Observing Figure 3, the intensity of the alkyd resin bands tended to increase once the samples were exposed to gas ageing. This behaviour was primarily from the hydrolytic reactions involving the binder functional groups; in fact, the ortho-phthalate ester group is sensitive to high humidity conditions, especially if acidified. The result was the decrease in the intensity of the aromatic C–H out-of-plane bending (741–711 cm^−1^) and the increase of the C–O–C stretching band at 1258 cm^−1^ of the phthalic group, respectively. However, the most evident degradation phenomenon was the increase in intensity and widening of the band at 1719 cm^−1^ related to the stretching of the carbonyl group and the formation of a defined shoulder at 1637 cm^−1^. This indicates the formation of oxidation reactions caused by ozone ageing, which leads to the presence of alcohols and carbonyl species [49]. Furthermore, the widening and increase of the band at 1637 cm^−1^ may also be due to the C=O stretching vibration in the diketones and their enol form. Finally, a decrease in C–H methylene absorptions at 2926 and 2855 cm^−1^ was observed due to the oxidation of the double bonds [7].

Figure 1 depicts the TIC after THM–SS-Py–GC/MS analysis of the alkyd resin exposed to ozone degradation at 50%RH and 80%RH. From a qualitative point of view, the comparison of the three different sets of samples showed that even after ozone ageing, there were no significant changes in Alk_80%RHO_3_. The situation was different in Alk_50%RHO_3_, as there was a partial increase of BA, and a drastic decrease of PE (in the form of tetra-PE methyl ester) and of some additives (in particular phthalates and butanal dimethylhydrazone).

As seen in Table 4, the main compounds detected in the Alk_ref, Alk_50%RHO_3_, and Alk_80%RHO_3_ set of samples were quantitatively compared. As qualitatively seen, PE varied from 29% for Alk_ref to 17% for Alk_50%RHO_3_, whereas for Alk_80%RHO_3_, the value was similar (27%). These data suggest that at 50%RH, the polyol was more subjected to oxidation. In the literature, pentaerythritol oxidation through ozonolysis and UV radiation is reported even in mild conditions, leading to the formation of ketones and aldehydes [50]. Unfortunately, the TMAH derivatization and THM procedure did not allow us to see these oxidation by-products. In addition, the effect of ozone is a phenomenon that occurs predominantly at the surface level, whereas PY–GC/MS analyses give insights from the bulk of the painted layers. However, their presence could explain the formation of a shoulder around 1637 cm^−1^ seen in the IR spectrum shown in Figure 3. Table 5 reports all the calculated molar ratios for the studied alkyd paints.

Regarding the lipidic fraction and, in particular, the presence of unsaturated fatty acids (oleic and linoleic acids), it would be expected that ozone ageing causes strong oxidation, with a consistent increase in oxidation products (primarily of dicarboxylic fatty acids, such as azelaic, suberic, and sebacic) and in A/P, D/P, and %D molar ratios, mainly observed in traditional and commercial oil paints. However, in this specific case, there was only a minimal increase in %D. The O/S ratio, index of oil maturity (Table 5), actually decreased in Alk_50%RHO_3_, but the C=C unsaturations were still present in similar amounts in Alk_ref. The presence of several oxidised octadecanoic acids (oxo-, hydroxy-, and methoxy-octadecanoic acids) produced by the oxidative scission of unsaturated fatty acids was significant in the ozone-aged samples.

The ratio of polyol and oil (PE/oil, Table 5) was very similar for Alk_ref and Alk_80%RHO_3_ (0.99 and 0.86, respectively), whereas it was halved in the case of Alk_50%RHO_3_ (0.46); this again indicates that PE was the most attacked fraction by induced degradation. Looking at the polycondensation scheme (Figure 2), it could be supposed that O_3_ acted on the OH group of the polyol with greater ease, also given the different energy strengths of OH and C=C bonds (respectively 460 and 680 kJ/mol). 

Moreover, the % increase of BA in the case of Alk_50%RHO_3_ could indicate that after such accelerated ageing, the thermal stability of the resin was reduced, as also reported in the case of photo-degradative treatment by solar radiation [8,9]. For the considerations made above for PE, the BA/PE ratio for Alk_50%RHO_3_ was doubled compared to the other two values (see Table 5).

### 3.2. Alkyd Paints

The presence of metal ions from pigments might act as a catalyst for polymerization and oxidation reactions, especially on the lipidic component of alkyd films [51,52]. Looking at the TIC obtained after PY–GC/MS analysis of alkyd paints (Table 5), it could be noticed that in general:The addition of pigments favoured the maturity of the oil; with a specific decrease in oleic acid unsaturations, the O/S values tended to decrease compared to the pure alkyd resin. This observation was particularly evident for PG18, where the O/S value was 0.97. In other cases, such as PB29, the O/S value remained similar to the pure resin.Dicarboxylic fatty acids (tertiary oxidation products in the oxidative polymerization of drying oils) increased, and consequently so did the values of the molar ratios A/P and D/P, as well as %D. The %D was higher in the cases of PG50, PY37, and PW6 paints. As a result, PhA/A values slightly decreased. On the contrary, PhA/PE, BA/PE, and polyol/oil (PE/oil) ratios were quite similar to the value of the pure resin.In paints containing PY37, PG50, and PB29, immediately after the phthalic acid RT, a peak was detected that was not present in pure alkyd and not even in other pigments. It had a structure similar to lactones that was not identified with certainty, but that could be related to the catalysing effect of these pigments during drying and curing (Appendix A).

When the samples were exposed to ozone ageing, the degradation reactions described for pure alkyd resin were influenced by the presence of inorganic pigments in the mixtures. In fact, through their physical–chemical properties, they could promote or reduce the degradation effects caused by pollutants. Considering that the increase and widening of the band at 1719 cm^−1^ and the presence of the shoulder at 1637 cm^−1^ were the main and more evident oxidation parameters observed in the spectra, they were be used to evaluate the stability of the alkyd binder according to the different inorganic pigments in paints. Figure 4 and Table 6 show the trends and area values obtained from the integrated FTIR absorbance band as a function of the relative humidity value used to age the paints studied.

The FTIR semi-quantitative evaluation obtained by integrating the absorbance band at 1719 cm^−1^ allowed for the confirmation that the gas ageing carried out at 50%RH was the most damaging for alkyd paints. The increase in intensity of this spectral signal and the decrease in the band to 741–711 cm^−1^ belonging to the aromatic CH out-of-plane bending of the phthalic group was due to the strong hydrolytic degradation. The reactivity of the ozone caused the breaking of the aliphatic and aromatic double bonds or the saturated aliphatic chains, thus leading to the decrease of the signal of the phthalic group, the widening of the band to 1719 cm^−1^, and the growth of a shoulder around 1637 cm^−1^. The latter could be related to the formation of carbonyls and carboxylic acids. As confirmed by a parallel study [53], this phenomenon could be explained by the intrinsic properties of water in humidified form once deposited on the polymeric surface, by the reactivity of ozone (see Section 3.3, General Discussion), and by the chemical properties of inorganic pigments in mixture. Indeed, mainly PB29, PW6, and PR101 mixed with the binder increased these degradation reactions with both relative humidity contents (Appendix A). The only exception, with gas ageing at 50%RH, was the alkyd paint with PY37, which caused the most significant increase in the intensity of the band at 1719 cm^−1^ and the formation of a band at 1640 cm^−1^ (no longer just a shoulder as previously seen). The degradation of the alkyd resin promoted by PY37 was probably due to the presence of Cd and S complexes in the pigment formulation that also increased the intensity of the band at 1115 cm^−1^, relative to the formation of the S=O species [49]. Unfortunately, this last consideration could not be confirmed as, although this increase was detectable, it was overlapped by the C–O stretching of the phthalic group. 

As seen for pure alkyd resin, data from bulk PY–GC/MS analysis showed that alkyd paints aged with ozone at 80%RH had similar behaviour to the corresponding reference alkyd paints (see Table 5). 

It should be noted that the benzoic acid decreased drastically in the sample Alk_PR101_80%RHO_3_, resulting in a BA/P value of 0.52. It was noted that another peak of benzoic acid was formed at 5.38 min, just before the peak considered in this study.

As reported in Korpany et al. [54], FeOOH (the main constituent of synthetic red iron oxide) could catalyse the oxidation and the dissociation of benzoic acid (Appendix A). Although in a minor amount, the same effect occurred with the pigment PB35 (stannate cobalt-based blue pigment), since cobalt oxide may act similarly to red oxide [55].

As clarified by the IR analysis on the surface, ageing with ozone at 50%RH showed the most significant degradation effects of alkyd paints; however, it was also confirmed by the Py–GC/MS analysis considering the bulk matrix. In fact, the Py–GC/MS results of Alk_50%RHO_3_ mixed with pigments showed the presence of many oxo-, methoxy-, and epoxy-octadecanoic acids, a lower content of oleic acid, an increase in A/P, D/P, and %D, and finally, a significant decrease in PE.

### 3.3. General Discussion

A very interesting trend is the fact that alkyd resin appears to be more susceptible to ozone oxidation with a relative humidity content of 50%, followed by 80%. This phenomenon is contrary to the effect of other corrosive gases in contact with this material, where an increase in RH is directly related to an increased corrosion rate. According to the literature [56,57], bulk liquid water properties occur when three or more monolayers of water are deposited on a surface. This occurs when, as tested in this study, a relative humidity value greater than or equal to 80% is employed. Thus, the aqueous layer accumulated at 80%RH on the resin surface has two effects: reducing both the intermediate reactions of ozone in the water film and the diffusion of the corrosive action of ozone on the organic surface. 

On the other hand, ageing the samples at 50%RH is equivalent to having two monolayers of water on the surface, making it a layered system. As a result, corrosion products from the water/ozone combination will be more concentrated and more in contact with the resin. This phenomenon will lead to a more significant degradation effect of alkyd paints when exposed to 50%RH, as demonstrated by the FTIR and Py–GC/MS results.

## 4. Conclusions

This study attempted to understand the effect on alkyd paints of ozone exposure at two different RH (50–80%RH). The employment of a comparative and systematic evaluation of FTIR and Py–GC/MS analysis, where the latter was used with specific experimental procedures and advanced semi-quantitative evaluations, allowed both the new formulations of alkyd paints and the effects of degradation induced by ozone to be clarified. Specifically, by comparing FTIR and Py–GC/MS results, the composition of Lukas commercial alkyd resin was characterized as an oil-modified polyester resin containing pentaerythritol (as polyol), ortho-phthalic acid (as polybasic acid), linseed oil (as drying oil), and benzoic acid (as the main additive, stopping agent). This formulation, already the object of studies on artists’ alkyds, differs from those studied in the last ten years and confirms the continuous variations in the selection of raw materials in complex painting mixtures. Some chemical changes in the unaged alkyd samples were observed by adding selected inorganic pigments, such as a decrease of the oleic acid unsaturations (especially in PG18) and an increase of dicarboxylic fatty acids, which may lead to different degradation behaviours of the alkyd matrix when exposed to ozone.

After ozone ageing, it was observed that the degradation reactions varied according to the relative humidity value (50 or 80%RH) used. Through surface-level spectroscopic analysis (FTIR-ATR), hydrolytic reactions were observed, mainly affecting the ortho-phthalic groups and leading to the formation of carbonyl species (1637 cm^−1^), especially at 50%RH. Bulk Py–GC/MS analyses highlighted that at 50%RH, the polyol (pentaerythritol) was the fraction most subject to oxidation. It is also interesting to note that the oil fraction was not so strongly affected by ozone ageing, except for the known oxidation of unsaturated fatty acids and the increase of dicarboxylic fatty acids). 

From the semi-quantitative FTIR and Py–GC/MS evaluations, it was observed that some pigments promoted these degradation reactions (i.e., PB29, PW6, and PR101) when the alkyd paint was aged at 50%RH. Moreover, PR101 was found to promote the dissociation of benzoic acid. 

From the conservation point of view, these results show that the systematic and combined use of various analytical techniques for studying commercial complex formulations, and the reactivity that the various pollutants have between them and the organic matrix are important factors in obtaining complete information on the degradation processes of these materials. Ozone is one of the most harmful atmospheric pollutants present in outdoor and museum environments; therefore, conservation practices such as the control of environmental parameters, the adequate arrangement of artwork during exhibitions, and the monitoring and prevention of their degradation conditions are evaluations to be considered to preserve the chemical–physical and aesthetic stability of modern and contemporary artwork.

In terms of future research, the technological development of advanced analytical techniques, the use of innovative methodologies and data evaluations, and the study of new artistic materials will allow an extensive and more detailed knowledge of polymeric film stability used in artwork to be obtained.

## Figures and Tables

**Figure 2 polymers-14-01831-f002:**
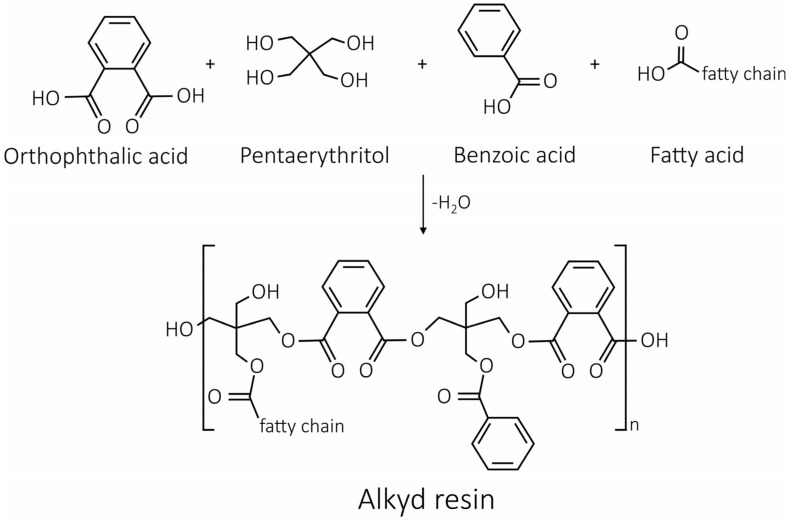
Schematic representation of the polycondensation reaction of alkyd resin identified from PY–GC/MS analysis.

**Figure 3 polymers-14-01831-f003:**
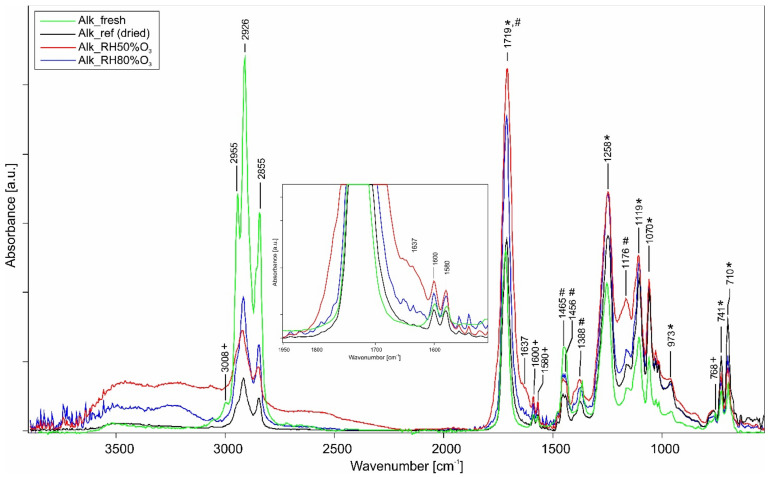
ATR-FTIR spectra of pure alkyd samples: comparison between fresh (green), pure alkyd reference (black), O_3_ + 50%RH (red), and O_3_ + 80%RH (blue) aged samples. The * marks all bands related to the phthalic-based compound, the + marks all bands related to phthalate, whereas the # indicates the oil bands.

**Figure 4 polymers-14-01831-f004:**
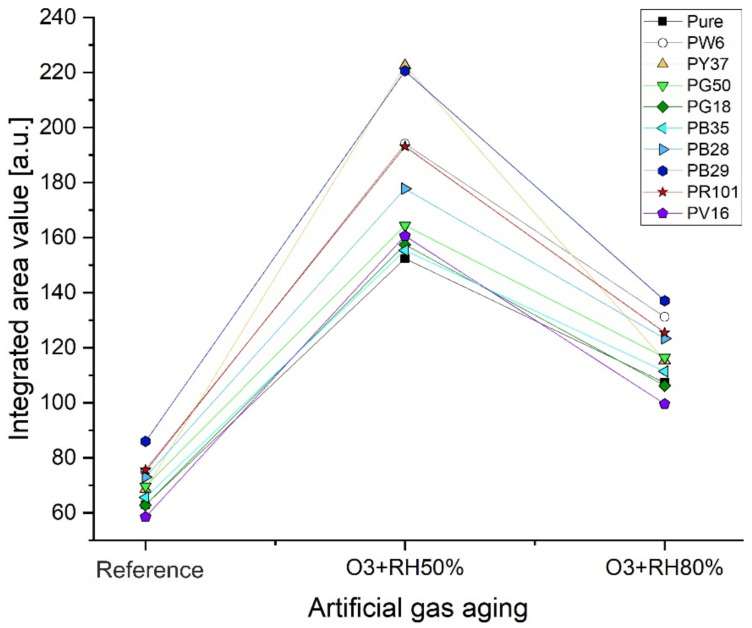
The trend of FTIR normalized and integrated absorbance band at 1719 cm^−1^ for the evaluation of alkyd resin stability when mixed with various inorganic pigments.

**Table 1 polymers-14-01831-t001:** List of materials analysed.

Binder	Pigments	Chemical Composition *	Colour Index (C.I.) Number
Alkyd resin (Alk) Commercial name: Alkyd Medium 4		Polymer oil-modified polyester-resin	
	Titanium white	TiO_2_	PW6
	Cadmium yellow	CdS	PY37
	Cobalt green	Co_2_TiO_4_	PG50
	Hydrated chromium oxide green	Cr_2_O_3_ · 2H_2_O	PG18
	Cerulean blue	CoSnO_3_	PB35
	Cobalt blue	CoO · Al_2_O_3_	PB28
	Artificial ultramarine blue	Na_8−10_Al_6_Si_6_O_29_S_2−4_	PB29
	Iron oxide red	Fe_2_O_3_	PR101
	Manganese violet	NH_4_MnP_2_O_7_	PV16

* Declared from the manufacturer.

**Table 2 polymers-14-01831-t002:** ATR-FTIR band assignment of materials investigated.

Material	Wavenumber (cm^−1^)	Absorption Band	Assignment
Alkyd resin	3008	Vinyl proton of C–H stretching	Phthalate plasticizer
	2926–2855	C–H stretching (sym–asym)	
	1719	C=O stretching	Oil and phthalic-based compound
	1600–1580	Aromatic ring C=C stretching	Phthalate plasticizer
	1465–1453	CH_2_ and CH_3_ bending (asym)	Oil
	1388	CH_3_ bending (sym)	Oil
	1258	C–O–C stretching (sym)	Phthalic-based compound
	1176	C–O stretching	Oil
	1119	C–O stretching (sym)	Phthalic-based compound
	1071	C–O stretching	Phthalic-based compound
	973	Out-of-plane CH deformation	Phthalic-based compound
	768	Aromatic C–H out-of-plane bending	Phthalate plasticizer
	741–711	Aromatic C–H out-of-plane bending	Phthalic-based compound
Titanium white (PW6)	606–546	TiO_2_ vibrations	
Cadmium yellow (PY37)	/	Below detector cut-off	
Cobalt green (PG50)	602	Co–O vibrations	
Hydrated chromium oxide green (PG18)	546–484	Cr–O vibrations	
Cerulean blue (PB35)	553	Co–O vibrations	
Cobalt blue (PB28)	641–553–486	Al–O and Co–O vibrations	
Artificial ultramarine blue (PB29)	1024–976	Al,Si–O_4_ asymmetric stretching	
Iron oxide red (PR101)	544–481	Fe–O vibrations	
Manganese violet (PV16)	3213–3068	O–H in the mineral	
	1416	[PO_4_]^3–^ vibrations	
	1032–995–905	P–O stretching (asym)	
	638–591–564–490	O–P–O bending	

**Table 3 polymers-14-01831-t003:** Pyrolysis fragments of Alk_ref detected by Py–GC/MS analysis.

Retention Time (min)	Compounds *	M^+^ (*m*/*z*)	Origin ^§^
4.999	6-Heptenoic acid ME	142 (74, 41, 43)	Monobasic acids (oil)
5.067	Heptanoic acid ME	144 (74, 87, 43)	Monobasic acids (oil)
5.724	Benzoic acid ME	136 (105, 77, 51)	Stopping agent (BA)
5.768	7-Octenoic acid ME	156 (55, 74, 43)	Monobasic acids (oil)
5.836	Caprylic acid ME	158 (74, 87, 43)	Monobasic acids (oil)
5.870	Pentaerythritol tetraME	128 (75, 45, 71)	Polyol (PE)
5.975	Pentaerythritol diME	131 (45, 71, 99)	Polyol (PE)
6.418	Pentaerythritol triME	178 (45,75, 71)	Polyol (PE)
6.642	Nonanoic acid ME	172 (74, 87, 55)	Monobasic acids (oil)
6.751	4-Methyl-2-piperidone	113 (42, 55, 69)	Additive (paint stabiliser)
7.679	8-Methoxyoctanoic acid ME	188 (45, 74, 124)	Monobasic acids (oil)
7.856	Phthalo lactone	134 (105, 77, 134)	Additive
8.026	1-Tetradecene	196 (43, 55, 57)	Additive
8.227	10-Undecenoic acid ME	198 (74, 55, 87)	Monobasic acids (oil)
8.394	Nonanoic acid, 9-oxo ME	186 (74, 87, 55)	Monobasic acids (oil)
8.475	Suberic acid diME	202 (129, 138, 74)	Monobasic acids (oil)
8.710	Phthalic acid diME	194 (163, 77, 76)	Polybasic acid (PA)
9.291	Azelaic acid diME	216 (152, 55, 74)	Monobasic acids (oil)
10.064	Sebacic acid diME	230 (55, 74, 125)	Monobasic acids (oil)
10.427	Allyl methyl phthalate	220 (163, 164, 104)	Additive
10.774	Butanal, dimethylhydrazone	114 (44, 85, 42)	Additive
11.866	2-Methoxyethyl methyl phthalate	238 (163, 58, 77)	Additive (plasticiser)
12.046	Palmitic acid ME	270 (74, 87, 43)	Monobasic acids (oil)
13.264	Oleic acid ME	296 (55, 69, 74)	Monobasic acids (oil)
13.427	Stearic Acid ME	298 (74, 87, 43)	Monobasic acids (oil)
13.594	Phthalic acid, furfuryl hexyl ester	334 (71, 149, 84)	Additive (plasticiser)
13.747	Methyl nonyl phthalate	306 (163, 149, 181)	Additive (plasticiser)
13.883	Methyl octyl phthalate	292 (163, 149, 181)	Additive (plasticiser)
14.203	Nonadecanoic acid ME I.S.	312 (74, 87, 43)	Internal standard
14.403	Linoleic acid ME	294 (67, 81, 95)	Monobasic acids (oil)
14.516	Methyl 4-methylpentan-2-yl phthalate	264 (163, 149, 181)	Additive (plasticiser)
14.665	Oxiraneoctanoic acid, 3-octyl ME	312 (155, 55, 41)	Monobasic acids (oil)
14.784	Oxiraneoctanoic acid, 3-octyl-, ME, cis-	312 (55, 74, 155)	Monobasic acids (oil)
14.869	Octadecanoic acid, 10-oxo- ME	312 (55, 43, 57)	Monobasic acids (oil)
15.029	Arachidic Acid ME	326 (74, 87, 43)	Monobasic acids (oil)
15.264	Hexadecanoic acid, 9,10,16-trimethoxy ME	360 (71, 95, 201)	Monobasic acids (oil)
15.907	Octadecanoic acid, 9,10-dihydroxy-, ME	330 (155, 55, 41)	Monobasic acids (oil)
16.907	Behenic acid ME	354 (74, 87, 43)	Monobasic acids (oil)

* ME = methyl ester, I.S. = internal standard; ^§^ BA = benzoic acid, PE = pentaerythritol.

**Table 4 polymers-14-01831-t004:** Normalized % in weight of the most abundant compound detected (fatty acids, polyol, polybasic acid, and stopping agent).

	Alk_Ref	Alk_50%RHO_3_	Alk_80%RHO_3_
Palmitic acid	9.01	11.66	9.12
Stearic acid	5.41	6.92	5.52
Azelaic acid	5.71	6.78	5.46
Suberic acid	1.19	2.73	1.15
Sebacic acid	0.67	0.72	0.67
Linoleic acid	0.36	0.94	0.98
Oleic acid	7.49	7.84	8.73
Pentaerythritol (di-, tri-, tetra-)	29.52	17.42	27.28
Phthalic acid diME	27.45	25.17	27.35
Benzoic acid ME	13.17	19.83	13.74

**Table 5 polymers-14-01831-t005:** Calculated molar ratios for pure alkyd paints, pigmented paints, unaged and aged with ozone, and 50% and 80%RH.

			Fatty Acids from Oil										Polyol/Oil
			P/S	A/P	D/P	O/S	%D	PhA/P	PhA/A	BA/P	PhA/PE	BA/PE	PE/Oil
Pure alkyd resin		Alk_ref	1.64	0.46	0.61	1.96	17.64	3.05	4.80	1.46	0.93	0.45	0.99
		Alk_50%RHO_3_	1.66	0.42	0.64	1.60	19.25	2.20	3.72	1.70	1.45	1.14	0.46
		Alk_80%RHO_3_	1.63	0.44	0.58	2.24	15.66	3.00	5.01	1.51	1.00	0.50	0.86
Alkyd paints	Ref.	PV16	1.69 ± 0.28	0.49	0.65	1.68	18.38	3.31	4.11	1.55	1.08	0.50	1.15
		PR101		0.40	0.64	1.63	22.93	2.98	5.14	1.72	1.11	0.64	1.06
		PB29		0.42	0.71	1.90	16.80	1.64	2.83	1.34	0.36	0.30	1.55
		PB35		0.49	0.65	1.57	19.18	2.41	5.09	1.44	0.85	0.51	1.12
		PB28		0.50	0.62	1.82	18.12	2.31	4.27	1.63	0.86	0.61	1.07
		PG18		0.49	0.61	0.97	19.99	2.25	4.29	1.76	0.77	0.60	1.33
		PG50		0.67	0.95	1.89	27.45	2.99	3.27	2.00	0.68	0.45	1.26
		PY37		0.77	0.74	1.76	20.57	1.35	2.09	1.57	0.35	0.41	1.26
		PW6		0.52	0.67	1.36	20.18	1.94	4.45	2.09	0.93	1.01	0.74
	50%RHO_3_	PV16	1.66 ± 0.32	0.63	0.73	0.86	25.89	2.47	2.84	2.07	1.76	1.12	0.52
		PR101		0.85	0.85	1.30	28.22	3.52	3.00	3.32	1.43	1.43	0.47
		PB29		0.63	0.70	1.40	23.27	3.64	4.21	4.08	1.34	1.62	0.39
		PB35		0.52	0.60	1.44	20.27	1.51	2.60	1.25	1.26	1.15	0.43
		PB28		0.48	0.58	0.54	18.23	2.91	4.44	2.38	1.41	1.62	0.51
		PG18		0.55	0.72	1.01	26.68	1.65	4.23	1.45	1.24	1.17	0.53
		PG50		0.70	0.82	1.59	29.45	2.23	3.52	1.90	1.32	1.32	0.45
		PY37		0.50	0.71	1.35	23.12	3.15	4.56	2.09	1.47	1.19	0.49
		PW6		0.60	0.70	0.51	24.64	2.38	2.89	0.52	1.39	1.23	0.50
	80%RHO_3_	PV16	1.63 ± 0.21	0.61	0.72	2.05	19.71	3.06	4.39	1.58	1.14	0,59	0,76
		PR101		0.43	0.68	1.70	16.52	2.38	2.89	0.52	4.35	2.2	0.29
		PB29		0.52	0.67	2.04	14.95	2.82	6.35	1.63	1.55	0.90	0.64
		PB35		0.54	0.76	1.94	21.25	2.20	3.66	0.86	1.24	0.48	0.56
		PB28		0.60	0.76	2.01	21.49	2.82	4.11	1.48	0.99	0.52	0.79
		PG18		0.49	0.64	0.93	22.98	2.08	3.92	1.61	1.47	1.14	0.54
		PG50		0.54	0.71	2.13	19.26	2.85	4.67	1.45	1.41	0.71	0.61
		PY37		05	0.54	1.74	15.06	1.99	4.90	1.99	0.95	0.96	0.84
		PW6		0.56	0.66	1.85	20.97	2.53	3.98	2.17	0.87	0.74	1.06

**Table 6 polymers-14-01831-t006:** Area values of the integrated band at 1719 cm^−1^ for all alkyd paint analysed.

	Reference	O_3_ + 50%RH	O_3_ + 80%RH
Pure alkyd	62.9	152.3	107.2
Alk_PW6	74.9	194.1	131.1
Alk_PY37	68.6	222.6	115.3
Alk_PG50	69.6	164.4	116.4
Alk_PG18	62.8	157.4	106.2
Alk_PB35	65.6	155.4	111.4
Alk_PB28	73.1	177.7	123.3
Alk_PB29	85.9	220.5	137.1
Alk_PR101	75.6	193.1	125.4
Alk_PV16	58.5	160.6	99.5

## Data Availability

Additional FTIR spectra, Py–GC/MS chromatograms, and information concerning methodology and semi-quantitative evaluations are available upon request.

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
