# Peer review of "How Can Ozone and Relative Humidity Affect Artists’ Alkyd Paints? A FT-IR and Py-GC/MS Systematic Study"

_polymers, 2022, doi:10.3390/polym14091831_

Round 1

Reviewer 1 Report

Dear authors. Congratulations. The paper, candidate, for its publishing in Polymers MDPI, scientifically sound good. Well detailed and "builded", the scientific interest is very appropiate and, also, its practical application for conservation heritage is achievable. I suggest that some minor corrections, particularlly about to english style, should be "polished".

Author Response

Response to Reviewer 1 Comments

Point 1: “…I suggest that some minor corrections, particularlly about to english style, should be "polished"”.

Response 1: We thank the Reviewer for the positive comments, the appreciation, and the interest shown in our study. The english language and style was carefully checked as required. 

Reviewer 2 Report

This research highlights the effect on alkyd paints of ozone exposure at two different RH (50%-80%RH). Alkyd paints are used for contemporary artworks. The analysis of the degradation of alkyd paints was done by FTIR and Py-GC/MS. The analysis result clarifies both the new formulations of alkyd paints and the effects of degradation induced by ozone. This formulation, already the object of studies on artists alkyds, differs from those studied in the last ten years and confirms the continuous variations in the selection of raw materials in complex painting mixtures. Ozone is one of the most harmful atmospheric pollutants present in the outdoor and museum environment, therefore, conservation practices such as the control of environmental parameters, the adequate arrangement of works of art during exhibitions, and the monitoring and prevention of their degradation conditions are essential to preserving the chemical-physical and aesthetic stability of modern and contemporary artworks. This research focuses on the study of new artistic materials that will allow obtaining an extensive and more detailed knowledge of polymeric films stability used in the art. The research is very interesting and maybe published after addressing the below issue.

In Figure 5, it is observed that the integrated area value is higher for O3+RH (50%) compared to the O3+RH (80%). The author needs the proper explanation to explain why lower RH is played a strong role in increasing integrated area value.

Author Response

Response to Reviewer 2 Comments
Point 1: “… In Figure 5, it is observed that the integrated area value is higher for O3+RH (50%) compared to the O3+RH (80%). The author needs the proper explanation to explain why lower RH is played a strong role in increasing integrated area value”.

Response 1: We would like to thank the reviewer for the appreciation and interest shown in our study. We agree with the reviewer's comment and on p. 13 a paragraph relating to the explanation of the higher integrated area value for samples aged at 50% RH was added. We hope that the clarification is exhaustive. We thank the reviewer for this careful observation.